# Why do small language models underperform? Studying LM Saturation via the Softmax Bottleneck

**Nathan Godey**[1,2], **Éric de la Clergerie**[1] **& Benoît Sagot**[1]
[1] Inria Paris, [2] Sorbonne Université
Paris, France
`nathan.godey@inria.fr`

## Abstract

Recent advances in language modeling consist in pretraining highly parameterized neural networks on extremely large web-mined text corpora. Training and inference with such models can be costly in practice, which incentivizes the use of smaller counterparts. However, it has been observed that smaller models can suffer from saturation, characterized as a drop in performance at some advanced point in training followed by a plateau. In this paper, we find that such saturation can be explained by a mismatch between the hidden dimension of smaller models and the high rank of the target contextual probability distribution. This mismatch affects the performance of the linear prediction head used in such models through the well-known softmax bottleneck phenomenon. We measure the effect of the softmax bottleneck in various settings and estimate that models based on less than roughly 1000 hidden dimensions tend to adopt degenerate latent representations in late pretraining, which leads to reduced evaluation performance.

## 1 Introduction

The representation degeneration problem is a common phenomenon that affects self-supervised learning methods used for textual data (Gao et al., 2019; Lai et al., 2023), among other modalities (Jing et al., 2022; Godey et al., 2024). This phenomenon consists in the emergence of degenerated structures in the intermediate latent spaces of language models throughout training. In particular, many observations on the intermediate representations of Language Models (LMs) have shed light on their low angular variability (or *anisotropy*) by showing that cosine-similarity between pairs of intermediate embeddings tend to be unexpectedly high (Zhou et al., 2021; Rajaee & Pilehvar, 2022). Other works have identified outlier dimensions that emerged during training (Puccetti et al., 2022). However, these observations were mostly made on relatively small-scale models of dimensions comparable to BERT (Devlin et al., 2019) or models from the GPT-2 family (Radford et al., 2019).

These models are usually composed of a neural network $f_\theta$ that takes sequences of tokens $(y_{<i}) \in [1, V]^{i-1}$ as inputs and produces a relatively low-dimensional contextual representation in $\mathbb{R}^d$, where $d$ is the *hidden dimension* of the model. They then rely on a *language modeling head* that produces logits for contextual token probabilities. A common choice for the language modeling head is a linear layer with parameter $W \in \mathbb{R}^{V \times d}$, where $V$ is the number of possible tokens. The resulting next-token probability distribution is then given by:

$$p(y_i) = \sigma(W f_\theta(y_{<i}))$$

where $\sigma$ is the softmax function.

In the language modeling field, the current trend consists in scaling up the generative pretraining approach introduced with GPT-2, which implies training neural models made of several billions of parameters on gigantic web-mined text corpora (Brown et al., 2020; Touvron et al., 2023; Almazrouei et al., 2023; Jiang et al., 2023). However, training and

serving such highly parameterized models raises energy and hardware-related problematics, which motivates for looking into achieving similar performance levels with smaller models (Sardana & Frankle, 2023).

Nevertheless, the evaluation of the Pythia model suite (Biderman et al., 2023) has shown that training small models on very large corpora could lead to *saturation*, in the form of a performance degradation in late pretraining. In this paper, we explore this saturation phenomenon through the lens of representation degeneration, and find that both phenomena strongly correlate. We further demonstrate that representation degeneration strongly occurs in the language modeling head of small models, and we theoretically and empirically show how a linear language modeling head can represent a performance bottleneck for architectures based on small hidden dimensions.

Overall, our contributions can be summarized as:

- We characterize the performance saturation of small language models through evaluation and extrapolation of the scaling laws;

- We find that the representations of smaller models degenerate concurrently with this saturation. We shed light on *rank saturation*, i.e. the explosion of the entropy of singular value distributions of small LM prediction heads;

- We empirically verify that the rank of the target contextual distribution is usually high. Moreover, we observe that regardless of the expressiveness of the output representations of a model, a linear head $W$ substantially affects performance when $rank(W) < 1000$;

- We theoretically quantify the performance limitation induced by a low-rank linear language modeling head.

## 2 Related Works

**Small LMs & Saturation**  Biderman et al. (2023) train Pythia, a suite of models of various sizes on 300B tokens from the Pile (Gao et al., 2020), and release the weights for an exhaustive number of checkpoints during training. They notice that smaller models suffer a performance decrease on the Lambada dataset (Paperno et al., 2016) in late training. The scaling laws (Kaplan et al., 2020; Hoffmann et al., 2022) predict that training smaller models on large corpora is suboptimal in terms of compute. However, recent initiatives (Zhang et al., 2024; Faysse et al., 2024; Team et al., 2024) have pretrained smaller language models on large datasets, motivated by inference cost reduction (Sardana & Frankle, 2023).

**Softmax Bottleneck**  The concept of *softmax bottleneck* was introduced in Yang et al. (2018), where the authors show that a model using a hidden dimension inferior to the rank of the contextual probability matrix cannot predict correctly in every context. They then hypothesize that this rank is relatively high in natural language and propose an alternative method for the predictive layer of language models. Subsequent works have explored negative effects of the softmax linear layer on language modeling performance (Chang & McCallum, 2022) and possible alternatives (Lin, 2021; Kanai et al., 2018). We extend this line of work by quantifying the critical dimensionalities involved in the softmax bottleneck.

**Representation Degeneration**  is a phenomenon in which pretrained models tend to adopt low-entropy singular value distributions (Jing et al., 2022). In language modeling, representation degeneration takes the form of anisotropy (Ethayarajh, 2019; Rajaee & Pilehvar, 2021) and was proven to be related with the Zipfian shape of token distribution (Gao et al., 2019; Biś et al., 2021). We study this phenomenon along training and its relation with saturation.

**Data Dimensionality and Performance**  Sharma & Kaplan (2022) link the scaling laws observed across pretrained models to data dimensionality, through the lens of Intrinsic Dimension (Camastra & Staiano, 2016). While they show that Singular Value Decomposition (SVD) is not suited for studying the dimensionality of the data manifold in the universal

approximation paradigm, we argue that it is well-suited, to a certain extent, when studying the performance of a linear classifier limited by the dimensionality of input representations.

## 3  Language Model Saturation

We first verify that we can indeed observe and quantify performance saturation for the Pythia checkpoints, as they are the only released intermediate checkpoints for a wide range of model sizes. We measure the cross-entropy of Pythia checkpoints on 50k tokens randomly sampled from their pretraining dataset, i.e. The Pile (Gao et al., 2020).

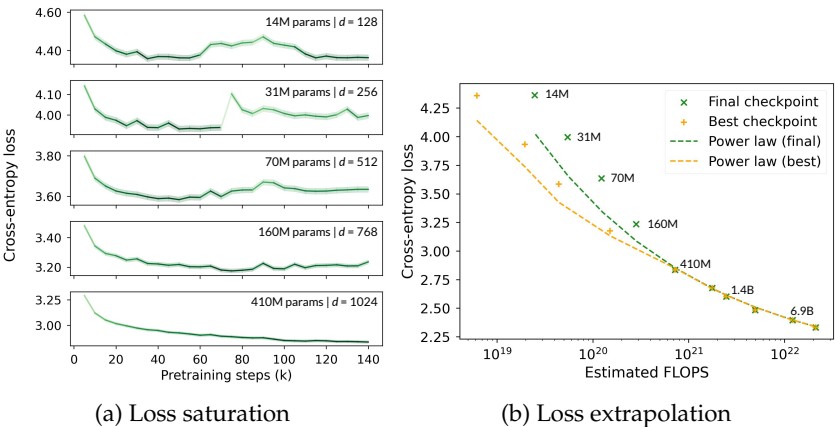

(a) Loss saturation                     (b) Loss extrapolation

Figure 1: Performance of Pythia models on the Pile. On the left, we compare training dynamics of models from 14M (top) to 410M (bottom) parameters, displaying darker shades as we approach the minimal value. On the right, we fit a power law on larger models and find that final checkpoints of smaller models underperform compared to predictions.

In Figure 1a, we clearly see that models up to 410M parameters suffer from the saturation phenomenon, characterized as an increase of the in-domain loss in advanced training stages.

In Figure 1b, we fit a scaling law in the style of Hoffmann et al. (2022) on data points from models ranging from 410M parameters, only optimizing for model-related constants ($A$ and $\alpha$) while reusing all other values ($B = 410.7$, $\beta = 0.28$, $E = 1.69$). We recall the relation between parameter count $N$ and token count $T$ given in Hoffmann et al. (2022):

$$L(N, T) = \frac{A}{N^\alpha} + \frac{B}{T^\beta} + E$$

We find that optimal parameters are $A = 119.09$ and $\alpha = 0.246$. We display the fitted curves for token counts that correspond to best and final checkpoints. We observe that the final checkpoints underperform the extrapolation by 8% in average. The loss-minimizing (*best*) checkpoints, which are expected to fall short of the extrapolation due to their incomplete learning rate cooldown, only underperform it by roughly 4%.

A similar performance saturation is also observed on datasets used for evaluation in the LM Evaluation Harness (Gao et al., 2023), as shown in Table 1.

| Checkpoint | Lambada (ppl.) ↓ | Lambada ↑ | StoryCloze ↑ | WikiText (ppl.) ↓ | SciQ ↑ | ARC-e ↑ |
|---|---|---|---|---|---|---|
| Best | **24.6** | **40.3** | **59.6** | **30.47** | **79.6** | **46.5** |
| Final | 32.9 | 38 | 57.2 | 33.4 | 73.4 | 43.2 |

Table 1: Zero-shot performance of Pythia-160M best and final checkpoints on evaluation datasets. Unless specified, we report accuracy for all tasks.

# 4 Performance Saturation is Rank Saturation

## 4.1 Anisotropy at Scale

Anisotropy is a common form of representation degeneration that has been observed among various small language models. It consists in reduced angular variability of the representation distribution at a given layer. Previous works (Ethayarajh, 2019; Godey et al., 2024) notice that almost all layers of small Transformers language models are anisotropic. A common measure for anisotropy in a set $H$ of vector representations is the average cosine-similarity:

$$\mathcal{A}(H) = \frac{1}{|H|^2 - |H|} \sum_{h_i, h_j \in H, i \neq j} \frac{h_i^T h_j}{||h_i||_2 \cdot ||h_j||_2}$$

However, it remains unclear whether anisotropy affects models with over 1 billion parameters. In order to address this question, we compute average cosine-similarity of intermediate representations across layers in suites of models; namely GPT-2 (Radford et al., 2019), OPT (Zhang et al., 2022), Pythia (Biderman et al., 2023), and Gemma (Team et al., 2024). We use a subsample of The Pile (Gao et al., 2020), as we hypothesize that the domain of this dataset includes or matches the domain of the pretraining datasets used in these suites.

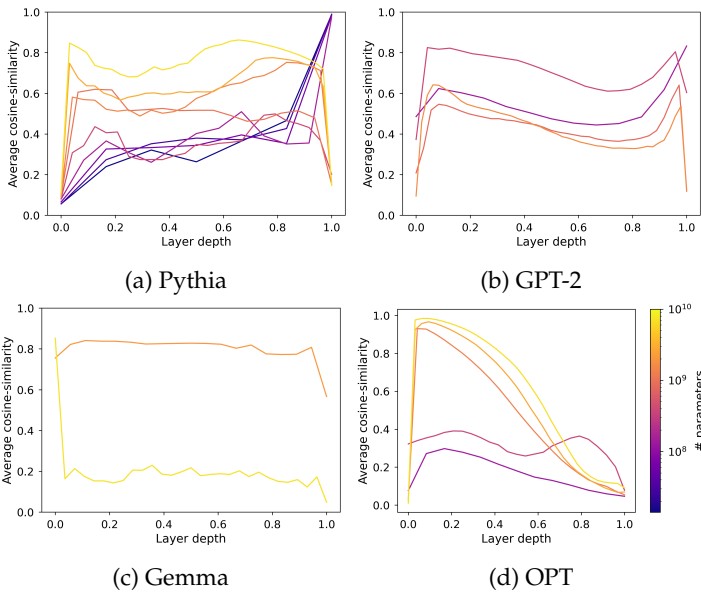

Figure 2: Anisotropy in function of layer depth (i.e. order in the forward pass).

In Figure 2, we observe that most layers of Transformers models are anisotropic to some extent, regardless of the scale. Nevertheless, there seems to be a dichotomy in the last layer, where models are either nearly isotropic or highly anisotropic. Interestingly, we notice that the dichotomy aligns with the one of the saturation phenomenon for the Pythia suite, where only models containing 160M or fewer parameters seem affected by last-layer anisotropy.

We thus decide to study the training dynamics of anisotropy for the Pythia suite, and compare them with the saturation phenomenon in Figure 3.

Figure 3 illustrates a neat correlation between the emergence of the performance saturation phenomenon and the appearance of anisotropy in the last-layer representations of the models. It also shows that anisotropy increases abruptly around the saturation point during training. Moreover, we see here that on a specific in-domain corpus, the models quickly lose performance at saturation and never seem to fully recover from this explosion.

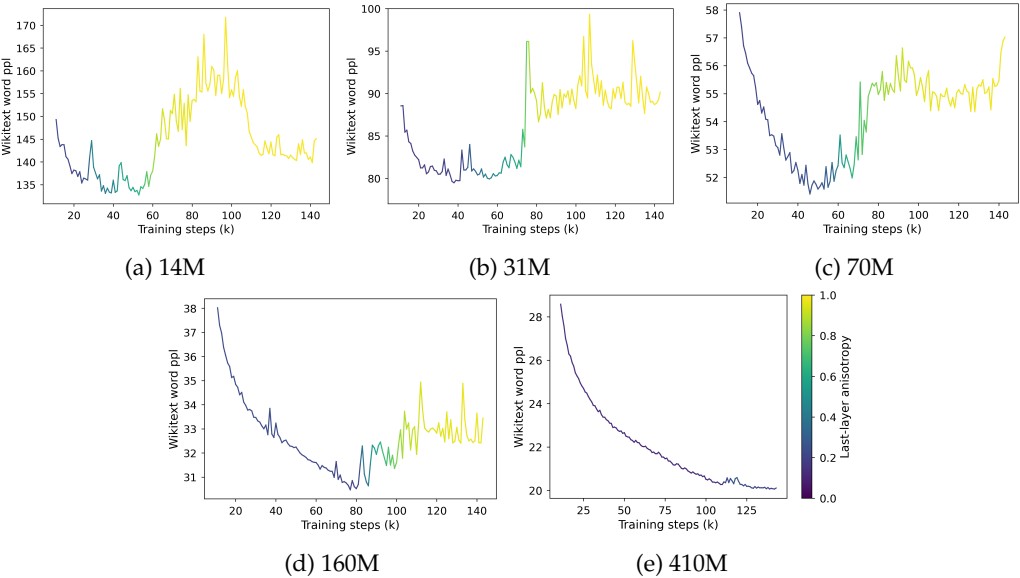

Figure 3: Evolution of the language modeling performance on the Wikipedia test set from the LM Evaluation Harness (Gao et al., 2023) and last-layer anisotropy of Pythia models along training (color).

## 4.2 Singular Values Saturation

Average cosine-similarity is a valuable measure of the uniformity of a distribution, but including other metrics can help to better capture the complexity of some manifolds (Rudman et al., 2022). Moreover, it only focuses on the output embeddings of the language models, and not on their weights. In this section, we extend our analysis by studying the singular value distributions of the language modeling heads, to link our empirical observations to our theoretical findings. In Figure 4, we display the singular value distributions of the final predictive layer weights $W$ along training.

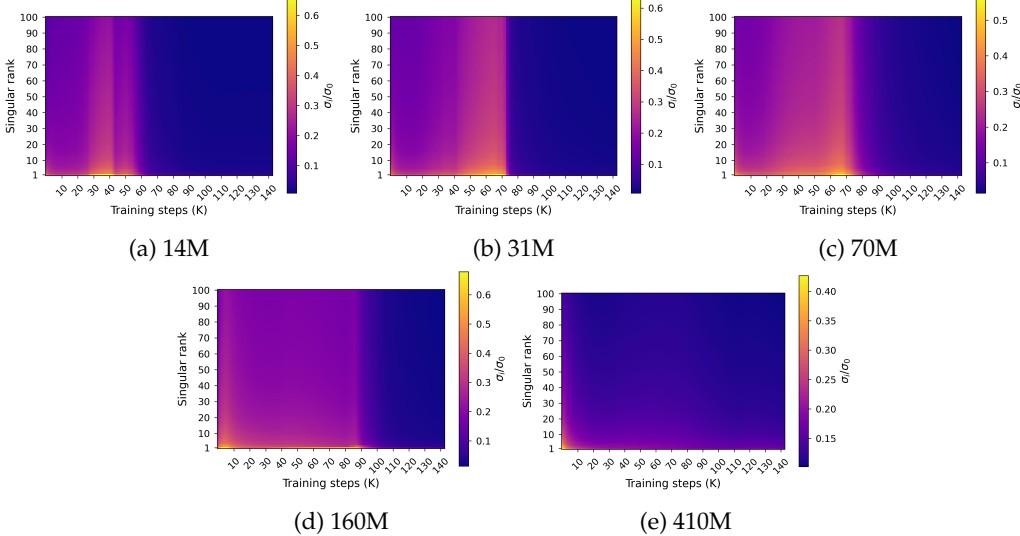

Figure 4: Evolution of the singular value distributions of the LM heads of Pythia models during training, normalized by the maximum singular value.

Figure 4 sheds light on a specific pattern of spectral saturation, roughly co-occurring with the performance saturation phenomenon. It shows that the singular value distribution progressively flattens during training, and nearly reaches uniformity before abruptly evolving towards a spiked distribution with a high maximal singular value, relatively to the other ones.

In order to quantify this behavior more accurately, we use a *singular entropy metric*, computed as the Kullback-Leibler divergence between the normalized singular value distribution and the uniform distribution.

Figure 5 shows that singular distributions evolve differently for models using less than 410M parameters than for the larger ones. The heads of small models see their singular value distributions become increasingly uniform, up to a point where they degenerate abruptly, which again correlates with the LM performance drop. The singular value distributions of larger models tend to be more stable, and do not display clear monotonic patterns throughout training.

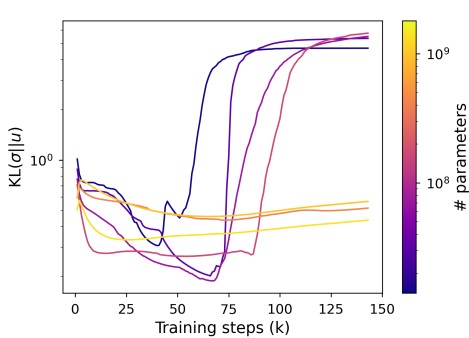

Figure 5: Training dynamics of the singular entropy, for different Pythia models.

## 5    The Softmax Bottleneck & Language Dimensionality

### 5.1    Inherent Dimensionality of Natural Language

Intuitively, the saturation of the singular values distribution observed only for smaller models in Section 4.2 questions the dimensionalities involved in the optimization of the LM head. In this section, we propose to empirically measure a critical value for the rank of the LM head, and to estimate the dimensionality of the contextual probability distribution the head's outputs are supposed to match.

In order to empirically measure the effect of the rank of the linear head, we propose to train rank-constrained heads on pretrained contextual representations from highly-parameterized language models. In order to control the maximum rank $r$, we consider heads of the form $W = AB \in \mathbb{R}^{V \times d}$, where the coefficients of $A \in \mathbb{R}^{V \times r}$ and $B \in \mathbb{R}^{r \times d}$ are drawn from $\mathcal{N}(0, 1)$ ($d$ being the hidden dimension of the model). The rank of such $W$ matrices is limited by the parameter $r \in [1, d]$, which we sweep over a wide range of values.

We freeze the language models and train the rank-constrained heads on their output representations on roughly 150M tokens, while adjusting the learning rate to the trainable parameter count (more details in Appendix B).

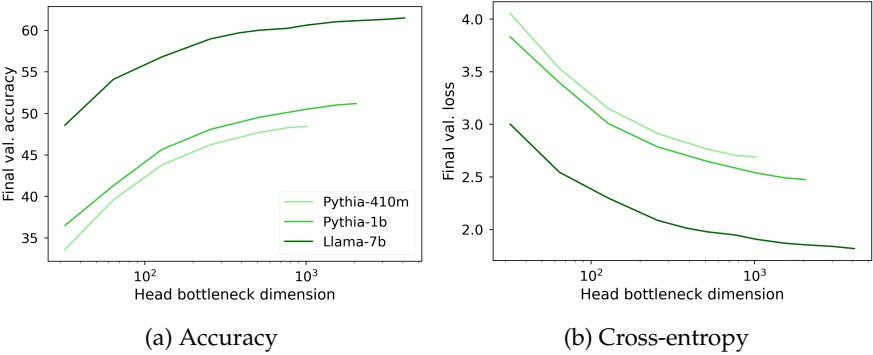

(a) Accuracy                                     (b) Cross-entropy

Figure 6: Performance of several models as the bottleneck dimension of the head increases.

In Figure 6, we observe that perplexity starts to noticeably decrease when the rank of the language modeling head $W$ is inferior to 1000, *regardless of the model size*. This hints that the head is not a major performance bottleneck for models with greater hidden dimensions, but that it may hurt performance for models with smaller ones independently of the quality of the output representations.

Another interesting factor to estimate is the dimensionality inherent to the data itself. To avoid possible effects related to specific inductive biases, we train naive 5-gram language models on several datasets of varying coverage (IMDb (Maas et al., 2011), Wikitext (Merity et al., 2016), and The Pile (Gao et al., 2020)), using two tokenizers of varying vocabulary sizes (30k tokens for Llama-2 and 50k tokens for Pythia). Given $C$ observed 5-grams, we consider the matrices $W \in \mathbb{R}^{C \times V}$ where each row is a probability distribution over possible tokens in a given 4-token context, and compute their singular value distributions, as in Terashima et al. (2003). In Figure 7, we report $W$-*error*, the minimal approximation error on $W$ for a matrix of rank $d$ as predicted by the Eckart-Young-Mirsky theorem (see Lemma 5.2), normalized by the Frobenius norm of $W$:

$$W\text{-error}(d) = \frac{||\sigma_{d+1:}||_2}{||W||_F}$$

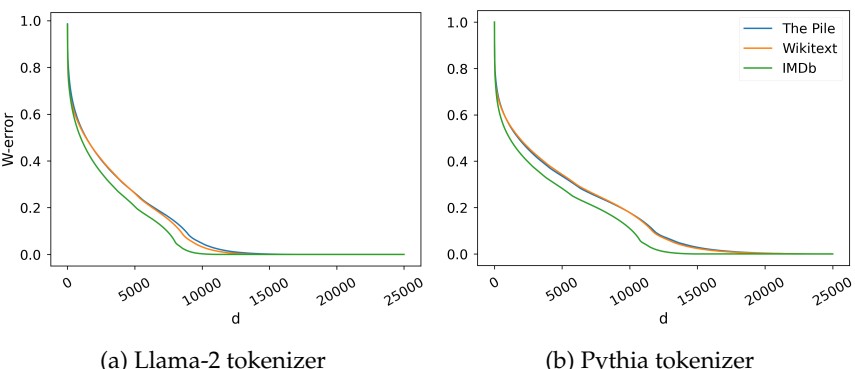

(a) Llama-2 tokenizer  (b) Pythia tokenizer

Figure 7: $W$-error as $d$ increases, for different tokenizers and datasets. We observe that while W-error can be halved using 1000 or 2000 dimensions, it only becomes negligible after 10,000-15,000 dimensions.

We find that the estimated rank of $W$ is non-negligible with respect to the usual magnitude of hidden dimensions. In the next section, we analyze the connection between the dimensionality of an ideal linear language modeling head and performance from a theoretical perspective.

## 5.2   A Theoretical Bottleneck

In this section, we aim at identifying a formal link between the inherent dimensionality of the contextual distribution and the performance bottleneck that can be attributed to the lower dimensionality of the output representations of a language model. To that end, we conceptualize a language modeling head optimized on *ideal* contextual representations, and we explore the relationship between its spectral properties and the performance gap induced when training a low-rank head on the same representations.

Let's consider a set $\mathcal{T}$ of sequences $(y_i)_{i \in [1, |y|]}$ of elements taken from a vocabulary of size $V$, representing the pretraining data. We consider a function $\phi^*$ that *perfectly* (e.g. in a bijective way) represents a given context $y_{<i}$ as a single real vector of *infinite* dimension. As we do not focus on $\phi^*$, we can simplify the notations by introducing the contextual representations $x_i^* = \phi^*(y_{<i})$.

The task of the linear language modeling head can be formalized as an optimization problem on the matrix $W$:

$$W^* = \underset{W \in \mathbb{R}^{V \times \infty}}{\operatorname{argmin}} \sum_{y \in \mathcal{T}} \sum_i \mathcal{L}(W, x_i^*, y_i) \tag{1}$$

where $\mathcal{L}$ is the cross-entropy objective defined using the softmax function $\sigma$ as:

$$\mathcal{L}(W, x, y) = -\log(\sigma(Wx)_y)$$

In practice, a neural language model $\phi_\theta$ produces contextual representations $x_i = \phi_\theta(y_{<i})$ of dimension $d \in \mathbb{N}^*$. The linear language modeling head $W_\theta \in \mathcal{R}^{V \times d}$ is trained concurrently with $\phi_\theta$ with the same objective as in Equation 1.

We focus on the maximal expressiveness of a lower-dimensional head: when provided with *perfect* contextual representations $x_i^*$, what is the maximal performance level of a linear language modeling head of maximal rank $d$? This question can be put in mathematical terms:

$$W_d^* = \underset{W \in \mathbb{R}^{V \times \infty}}{\operatorname{argmin}} \sum_{y \in \mathcal{T}} \sum_i \mathcal{L}(W, x_i^*, y_i) \text{ s.t. } rank(W) \leq d \tag{2}$$

Lemma 5.1 shows that by approaching $W^*$ directly, we can asymptotically expect to close the performance gap.

**Lemma 5.1** *(proof in Appendix A.1) Let's consider $W \in \mathbb{R}^{V \times \infty}, M \in \mathcal{H}^{V \times \infty}$ the matrix unit sphere for the Frobenius norm $||\cdot||_F$, and $\varepsilon \in \mathbb{R}_+^*$ such that $W = W^* + \varepsilon M$ . When $\epsilon \to 0$:*

$$|\mathcal{L}(W, x_i^*, y_i) - \mathcal{L}(W^*, x_i^*, y_i)| = O(\varepsilon)$$

Hence, our problem is linked to a low-rank matrix approximation (Kumar & Schneider, 2017), which has direct connections with spectral theory. In our case, we can use the Eckart–Young–Mirsky theorem.

**Lemma 5.2** *(Eckart–Young–Mirsky theorem) Let's consider $(\sigma_i)$ the singular values of $W^*$ in decreasing order, and $\mathcal{M}_d$ the set of matrices in $\mathbb{R}^{V \times \infty}$ of rank $d < V = rank(W^*)$. Then:*

$$\min_{W_d \in \mathcal{M}_d} ||W_d - W^*||_F = \sqrt{\sum_{i=d+1}^{V} \sigma_i^2}$$

Combining all of the above yields Theorem 5.3.

**Theorem 5.3** *(proof in Appendix A.2) Let's consider $(\sigma_i)$ the singular values of $W^*$ in decreasing order. Then, when $d \to V$, the loss gap induced by a d-dimensional bottleneck on the linear LM head follows:*

$$\sum_{y \in \mathcal{T}} \sum_i \mathcal{L}(W_d^*, x_i^*, y_i) - \mathcal{L}(W^*, x_i^*, y_i) = O\left(\sqrt{\sum_{i=d+1}^{V} \sigma_i^2}\right)$$

These properties shed light on how the dimensionality of the ideal language modeling head impacts the performance when the LM head is low-rank. However, the relation obtained in Theorem 5.3 is not particularly strong, as discussed in Appendix A.2.

In Figure 8, we compare the results of the head bottleneck experiment of the Pythia-1B model in Section 5.1 to the $W$-error on the head of the same model as the bottleneck dimension $d$ evolves. It shows that the loss gap grows slowly with the $W$-error, implying that even when the allowed rank would lead to a poor approximation of $W$, the performance can still remain acceptable. We notice that the performance starts decreasing when the $W$-error outgrows 0.6.

## 6 Discussion

One way to address the problem at hand could be to train shallow small language models, increasing hidden dimension at the expense of other hyperparameters, such as layer count or feed-forward dimension. However, we believe that such research directions may not be promising in this context. Previous works have extensively explored and optimized the hyperparameter choices for various architecture sizes. The impact of width and depth has been extensively studied (Merrill et al., 2022; Tay et al., 2022; Petty et al., 2023), often showcasing the importance of depth in final performance and generalization capabilities.

Another possible way forward consists in implementing more expressive softmax alternatives (Yang et al., 2018; Chang & McCallum, 2022) in the context of pretraining small language models on large datasets. We leave the exploration of such techniques for future work.

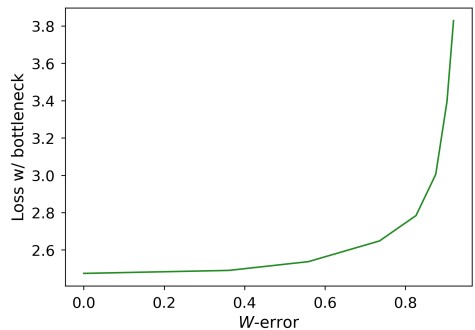

Figure 8: Final loss with trained rank-constrained heads (mimicking $W_d^*$), as a function of the theoretical $W$-error for rank $d$ on the head of the Pythia-1B model.

We also believe that further exploration of the specific nature of the singular components after the collapse we describe in Section 4.2 could improve our understanding of LM saturation. We hypothesize that the resulting dominating components are correlated with token frequency, based on previous works that link anisotropy with token frequency (Gao et al., 2019; Ethayarajh, 2019; Biś et al., 2021) and show the importance of token frequency in the LM head mechanism (Meister et al., 2023).

Beyond the scope of this article, we argue that our work demonstrates that last-layer anisotropy is symptomatic of performance saturation, and is thus likely not a desirable property of language models. We also advocate that this work paves the way towards a better understanding of the structure of the contextual probability distribution, which could also enhance our interpretation of the scaling laws.

## Conclusion

Small language models can be affected by performance saturation during training. We find that this phenomenon can be explained by an inherent difficulty in mapping a low-dimensional output representation space to a high-rank contextual probability distribution through a linear language modeling head. Indeed, we show a theoretical link between the performance gap induced by a smaller hidden dimension and the spectral properties of the contextual probability distribution.

We empirically confirm that the rank of such a mapping can be expected to be relatively high compared to regular hidden dimension choices. Moreover, we conduct experiments to measure the impact of constraining the rank of the LM head on the performance of a large language model. Our results show that performance noticeably drops when using a hidden dimension smaller than roughly 1000. We further analyze the saturation phenomenon through the lens of spectral analysis and find that the emergence of last-layer anisotropy that only affects small models can be correlated with saturation. We also show that the LM heads of small models concurrently suffer from *spectral* saturation, i.e. a uniformization of singular values that leads to a degenerated state.

Our work paves the way for a better understanding of the consequences of the softmax bottleneck on language modeling, and for the conception of language models that better embrace the complexity of the target probability distribution.

## Limitations

The main limitation of this article is the relatively small amount of saturated language models we studied. As it is the only suite of language models trained in the range of interest to release an extensive amount of intermediate checkpoints, we could only observe the training dynamics of small Pythia models. Although we observe strong last-layer anisotropy for the smallest GPT-2 model, we cannot tell with certainty whether it suffered from saturation. The OPT-125m model does not display a strong last-layer anisotropy, which could indicate that it was not affected by the saturation phenomenon.

Nevertheless, we argue that this paper does not show that *all* small models should suffer from saturation, but rather that the saturation of small language models is symptomatic of a limitation that may affect language models that are based on a relatively small hidden dimension. Furthermore, we do not state that there is a causality relationship between degeneration and low hidden dimension choices, but rather expose a strong correlation between both phenomenon that can be explained through the prism of our softmax bottleneck analysis.

Another limitation of this work is the loose nature of the mathematical connection that we establish between the dimensionality of the ideal language modeling head and the rank-constrained performance (cf. Theorem 5.3). Moreover, it can also be argued that considering *ideal* $x_i^*$ representations is an ill-defined notion. We argue that the reasoning behind Theorem 5.3 could be applied to any contextual representations, as the *ideal* nature of $x_i^*$ is not necessary in the demonstrations. The word *ideal* reflects that our observations hold for $x_i^*$ representations obtained from *any underlying model*, to an extent that depends on the structure that these representations impose on the $W^*$ matrix for a given training set $\mathcal{T}$.

## Acknowledgements

We thank Song Duong for carefully reviewing this article and for his valuable suggestions.

This work was funded by the last author's chair in the PRAIRIE institute funded by the French national agency ANR as part of the "Investissements d'avenir" programme under the reference ANR-19-P3IA-0001.

This work was granted access to the HPC resources of GENCI operated by IDRIS (allocation 2023-AD011013680R1).

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

# A Proofs

## A.1 Lemma 5.1

The proof is mainly based on calculations and limited development:

$$|\mathcal{L}(W, x_i^*, y_i) - \mathcal{L}(W^*, x_i^*, y_i)|$$

$$= \left| -\log \frac{\exp((Wx_i^*)_{y_i})}{\sum_{j \in V} \exp((Wx_i^*)_j)} + \log \frac{\exp((W^*x_i^*)_{y_i})}{\sum_{j \in V} \exp((W^*x_i^*)_j)} \right|$$

$$= \left| -(\varepsilon M x_i^*)_{y_i} + \log \frac{\sum_{j \in V} \exp((W^*x_i^*)_j) \exp((\varepsilon M x_i^*)_j)}{\sum_{j \in V} \exp((W^*x_i^*)_j)} \right|$$

$$= \left| -\varepsilon(Mx_i^*)_{y_i} + \log \left( 1 + \frac{\sum_{j \in V} \varepsilon \exp((Mx_i^*)_j)}{\sum_{j \in V} \exp((W^*x_i^*)_j)} + o(\varepsilon) \right) \right|$$

$$= \left| -\varepsilon(Mx_i^*)_{y_i} + \varepsilon \frac{\sum_{j \in V} \exp((Mx_i^*)_j)}{\sum_{j \in V} \exp((W^*x_i^*)_j)} \right| + o(\varepsilon)$$

$$= \varepsilon \left| -(Mx_i^*)_{y_i} + \frac{\sum_{j \in V} \exp((Mx_i^*)_j)}{\sum_{j \in V} \exp((W^*x_i^*)_j)} \right| + o(\varepsilon)$$

The continuous function $M \longrightarrow \left| -(Mx_i^*)_{y_i} + \frac{\sum_{j \in V} \exp((Mx_i^*)_j)}{\sum_{j \in V} \exp((W^*x_i^*)_j)} \right|$ is bounded on the compact matrix unit sphere (i.e. where $||M||_F = 1$), which ends the proof.

**Remark :** This result could also be proven using a differentiability argument, but we prefer to display a more precise relation between the loss gap and the error on the $W$ matrix approximation, stressing out its quasi-linear nature. This formulation will hopefully pave the way for further exploration of this relation in future works.

## A.2 Theorem 5.3

Let us note $W_d$ the best approximation of $W^*$ of rank $d$ with respect to the Frobenius norm. By definition of $W_d^*$, we have that:

$$\left| \sum_{y \in \mathcal{T}} \sum_i \mathcal{L}(W_d^*, x_i^*, y_i) - \mathcal{L}(W^*, x_i^*, y_i) \right| \leq \sum_{y \in \mathcal{T}} \sum_i |\mathcal{L}(W_d, x_i^*, y_i) - \mathcal{L}(W^*, x_i^*, y_i)| \quad (3)$$

The Eckart-Young-Mirsky theorem tells us that when $d \to V$,

$$||W_d - W^*||_F = \sqrt{\sum_{i=d+1}^{V} \sigma_i^2} \to 0$$

By defining $\varepsilon = W_d - W^*$, we can apply Lemma 5.1 and show that:

$$|\mathcal{L}(W_d, x_i^*, y_i) - \mathcal{L}(W^*, x_i^*, y_i)| = O(||W_d - W^*||_F) = O\left( \sqrt{\sum_{i=d+1}^{V} \sigma_i^2} \right)$$

From Equation (3), we have that:

$$\left| \sum_{y \in \mathcal{T}} \sum_i \mathcal{L}(W_d^*, x_i^*, y_i) - \mathcal{L}(W^*, x_i^*, y_i) \right| = O\left( \sqrt{\sum_{i=d+1}^{V} \sigma_i^2} \right)$$

By definition of $W^*$ and $W_d^*$, we also have that:

$$0 \leq \sum_{y \in \mathcal{T}} \sum_i \mathcal{L}(W_d^*, x_i^*, y_i) - \mathcal{L}(W^*, x_i^*, y_i) = \left| \sum_{y \in \mathcal{T}} \sum_i \mathcal{L}(W_d^*, x_i^*, y_i) - \mathcal{L}(W^*, x_i^*, y_i) \right|$$

which ends the proof.

**Remark** The bound used in Equation (3) can be rather loose in practice. We can think of no particular reason why approaching $W^*$ directly should be the optimal way to minimize the loss on $\mathcal{T}$. Hence, the presented result should be taken carefully, and we leave the refinement of such an analysis for future work.

## B Hyperparameters

### B.1 Constrained head experiments (Figure 6)

We freeze the pretrained weights in the Transformer layers, and we train each rank-constrained head (i.e. in the form $W = AB$ with $r$ as the inner dimension of the matrix product) for various values of $r$ on 150M tokens sampled from The Pile using 4 V100 GPUs for the Pythia models and 4 A100 GPUs for Llama-7B. We use the hyperparameters from Biderman et al. (2023), except for the batch size which we set to 256 as it fits our hardware setup better. As the trainable parameter count evolves with $r$, we search for the best-performing learning rates among values ranging from $1 \cdot 10^{-3}$ to $5 \cdot 10^{-2}$.

We report the chosen learning rates in Figure 9.

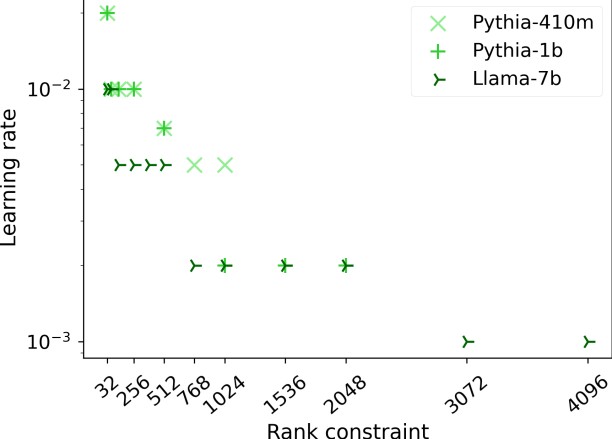

Figure 9: Chosen peak learning rates used for the rank-constrained head experiments for each model.