# OpenReview forum: "Why do small language models underperform? Studying Language Model Saturation via the Softmax Bottleneck"
_colmweb.org/COLM/2024/Conference — COLM_

### Official Review · Reviewer_gtfa · 2024-04-17

**Rating:** 9
**Confidence:** 4
**Ethics Flag:** 1

**Summary:**

The paper find that there are strong correlation between the following things
1. Loss saturation (i.e., instability of the loss after training for a long time)
2. Performance drop in the downstream tasks
3. Anisotropy distribution in the embedding space
4. Low effective dimension in the word embedding space (i.e., problem caused by softmax bottleneck)

**Questions To Authors:**

Small presentation suggestions:

1. The purpose of Figure 1 (b) is not very clear to me. Maybe you could explain more about what we can learn from this result.

2. When I first read Figure 7 and the definition of W-error(d), I don't understand what you tried to analyze. I understand it after reading Theorem 5.3. Thus, I suggest to move Figure 7 and its corresponding explanation to the end of section 5.2 (after your theoretical results).

**Reasons To Accept:**

Disclaimer: I did some research in the related topics previously, so I would like to admit that my evaluation might contain some positive biases toward this research direction. Too few papers like this one that studies these fundamental and practical problems. I believe this kind of study is very important and won't be outdated in the near future.

1. Overall, the presentation is clear and easy to read for me (maybe partially due to my prior knowledge in this field).
2. The correlations are very strong. I reviewed several analysis papers for language models before and it is very rare that you can observe such strong correlations.
3. The connection between loss saturation and anisotropy distribution / softmax bottleneck is a novel discover.
4. The authors designed several good experiments and theoretical analysis to quantify the seriousness of the problems.
5. Build the connections between several previous findings or research directions quantitatively, including softmax bottleneck, loss saturation, and anisotropy representation.

**Reasons To Reject:**

I think the rejection reasons are fairly minor to me. You don't have to spend time writing the rebuttal.

1. In the title, the why question seems to suggest this paper build causality relation between observations. It would be better to emphasize in the paper that correlation is not causality. In addition, it would be even better if the authors can find whether the loss saturation causes the anisotropy representation or the anisotropy representation causes loss saturation, and why.

2. Except for the connection between loss saturation and anisotropy distribution, the rest of the empirical and theoretical analyses is somewhat novel but not super novel (although showing them quantitatively is pretty useful and the execution is done very well).

3. I am not sure how easy to understand this paper for the researchers who do not study related problems before.

---

> ### Author Rebuttal · Authors · 2024-05-31
>
> We are truly grateful for the very positive feedback from reviewer gtfa.
>
> ## Reasons to Reject
> 1. We agree that the causality is not perfectly established in the paper, as it is very hard to derive clear causality from such highly-parameterized models for which reproducing pretraining is rather costly and conducting ablations is rather costly. We never explicitly mention such a causality, and describe correlations wherever required in the paper. Based on your comment, we will include a causality/correlation statement in the Discussion section of our final version.
> 3. As some other reviewers have pointed out, it may be tough for less knowledgeable readers to understand the paper as it is in its current form. As we described in the other answers, we have worked on a revised version of the paper in which the writing has been improved for publication to a broader audience.
>
> ## Questions
> 1. The purpose of Figure 1b is to show that if we extrapolate the expected performance of smaller Pythia models from the final performance of larger ones, we observe that models ranging from 14m to 160m parameters perform worse than expected. However, one could argue that extrapolating the larger models performance from the smaller models would lead to the opposite conclusion (i.e. large models underperform). Hence, we add another extrapolation (using the *best* checkpoints) to show that smaller models were closer to the performance predicted by the scaling laws mid-training, before the saturation phenomenon. We will add a sentence to make this clearer in the corresponding paragraph.
> 2. In our revised version, Figure 7 is after section 5.2.

---

> > ### Comment · Reviewer_gtfa · 2024-06-05
> > **Thanks for the response**
> >
> > I have read the rebuttal and have no further question.

---

### Official Review · Reviewer_vM6v · 2024-04-26

**Rating:** 5
**Confidence:** 3
**Ethics Flag:** 1

**Summary:**

The paper provides some experimental and theoretical evidence that the softmax bottleneck explains degenerate behaviour that has been observed when training LLMs with fewer parameters. In particular, it empirically shows that:
1. as the input dimension $d$ of the final classification layer shrinks, training loss tends to worsen and plateau late in training, and
2. as this happens, the average cosine similiarity between vectors in this layer suddenly spikes, and
3. singular value distributions also become suddenly high entropy (that this corresponds to 2 is not surprising), and
4. The the softmax bottleneck (where $d$ is much smaller than the number of classes $n$) theoretically relates to the observed problem with singular value distributions.

**Reasons To Accept:**

- I don't work on this problem, but my outsider view is that there is enough here that it will probably be of interest to those trying to figure out how to build effective compact models. I think the evidence presented here is circumstantial (there might be other ways to explain all of these things happening at once), but that seems ok from a scientific perspective; other people might be able to follow up by exploiting these observations or by coming up with better explanations for them.
- The figures are quite clear, and make good use of a color gradient to illuminate relationships between variables. Most of what I understood from the paper was due to the figures.

**Reasons To Reject:**

1. The writing of the paper is quite frankly a mess. It appears to have been written in a big hurry, and some parts are difficult to follow, especially at the beginning. For example, at first I thought I would follow the paper pretty well, having understood the abstract, and having myself worked on the softmax bottleneck and on training dynamics. But I found the first paragraph (and much of the introduction) to be utterly incomprehensible, winging about obscure terms like "The representation degeneration problem" and "low angular variability" as if everyone working on LMs knows what these are. I think I followed the paper pretty well after a few pages in, after the central ideas were explained and it became clear that these terms were just needless technical mumbo-jumbo. But this is a problem, because as it stands, I think most readers won't bother reading past the first page if the paper remains in its current form. I certainly wouldn't have if I hadn't been reviewing it. The proof is written as a bunch of math with no intuitive explanation, including of how it connects to the rest of the exploration; you have to squint a bit to see the relationship, or else take it on faith.

2. The paper makes an overly strong, unqualified claim that "a linear head W substantially affects performance when rank(W) < 1000". While this seems to be empirically true _for the models examined in this paper_ (which, as I understand, are all trained on the same data, only varying the number of parameters), I have a hard time believing that 1000 is some kind of magic number that is applicable in all circumstances. I would be surprised if the relevant threshold wasn't dependent on other factors like the size of the training data and vocabulary. I think this claim should be softened to account for these unknowns.

tl;dr: the paper exhibits a severe case of the Curse of Knowledge (https://en.wikipedia.org/wiki/Curse_of_knowledge). Fortunately this can be fixed with better writing. I really think a rewrite is in order before it is published.

---

> ### Author Rebuttal · Authors · 2024-05-31
>
> We warmly thank the reviewer for its thorough feedback.
>
> ### 1 - *The writing of the paper is quite frankly a mess*
> Based on the feedback of other reviewers, our understanding of the writing issue rightly identified by reviewer vM6v is that the paper is hard to follow *for readers that are less knowledgeable in this specific subfield*. We thus propose improvements for the camera-ready version that should make the article more inclusive:
> - **The representation degeneration problem** : We immediately provide two references that explore this specific problem in-depth, and we proceed to explain what it consists in. We will insert an explanation between the first two sentences of the papers, to make the connection between representation degeneration and anisotropy clearer: ”It [the rep. deg. problem] consists in the emergence of degenerated structures in the intermediate latent spaces of language models throughout training. In particular, many observations…”.
> - **Low angular variability** : We agree that readability could be improved by expanding the phrase, which we will do in the revised version: “...have shed light on their low angular variability, by showing that cosine-similarity between pairs of  intermediate embeddings tend to be unexpectedly high.”
> - In general, realizing that the page limit was greater than 8 (which we initially thought) let us reorganize and expand some paragraphs further. Although we cannot share the new version because of the conference guidelines, we obtained positive feedback from readers on the writing quality of this revision. The theoretical section was also reorganized and detailed in a new version of the article.
>
> ### 2. *I would be surprised if the relevant threshold wasn't dependent on other factors*
> The models we experimented with in Figure 6 are: the 410m and 1B models from the Pythia suite, which were trained on The Pile with a vocabulary of 50k tokens, and the 7B Llama-2 model, which was trained on another unspecified dataset using a vocabulary of 32k tokens. We chose The Pile for the low-rank head training as it is expected to match the pretraining dataset of Llama-2 closely (and exactly for the Pythia models).
>
> In all three cases, we see that the performance on The Pile validation data seems to noticeably deteriorate around a rank constraint of *roughly* 1000. Hence, we believe that the 1000 number is a good *estimation of the range* where the head rank starts being a significant in-domain performance bottleneck.

---

> > ### Comment · Reviewer_vM6v · 2024-06-04
> > **Thanks**
> >
> > I'm glad to hear that the writing of the paper has been improved.
> >
> > Also, thank you for the clarification about the training data and model sizes. However, I still don't think you can justify the framing of this as a constant on the basis of this handful of datapoints. Couching it as an _estimation of the range_ is an improvement over the strong claim in the paper (in the abstract no less). But even then, any such threshold is likely to be a function of other factors rather than a constant, and there isn't enough data in the paper to accurately determine what that function is, nor is it central to the main message of the paper. So I think this claim really ought to be softened and pointed to as a question for future work.

---

> > > ### Author Response · Authors · 2024-06-04
> > > **Response**
> > >
> > > We warmly thank the reviewer for engaging in the discussion process. We agree that this claim is strong as is, especially in the abstract. We can rephrase the sentence in the abstract as:
> > >
> > > "We measure the effect of the softmax bottleneck in various settings and *estimate* that models based on less than *roughly* 1000 hidden dimensions tend to adopt degenerate latent representations in late pretraining, which leads to reduced evaluation performance."
> > >
> > > We shall also modify the conclusion accordingly:
> > >
> > > "Our results show that performance *noticeably* drops when using a hidden dimension smaller than *roughly* 1000."
> > >
> > > Our goal is not to convey the idea that there is an exact threshold, but rather to show that there is a range of hidden dimension values for which the softmax bottleneck can be a noticeable limitation for language models, and that our empirical evidence suggests that this range of values extends up to *roughly* 1000 *in our experiments*.
> > >
> > > Finally, this claim is only made in the context of the paper, i.e. for causal language models trained with usual English large natural language datasets. Hence, the factors that could come into play are reasonably controlled : the training data is in large quantity and diversity, and the vocabularies extend from 30,000 to 150,000 tokens. We argue that our range estimate should approximately hold in these controlled setups.

---

### Official Review · Reviewer_sGwj · 2024-05-10

**Rating:** 6
**Confidence:** 4
**Ethics Flag:** 1

**Summary:**

The paper delves into a 'why' question concerning LM saturation through the softmax bottleneck. The key finding reveals that models with fewer than 1000 hidden dimensions, often representing the context, tend to adopt degenerate latent representations during late pretraining  characterised by a drop after a plateau, resulting in reduced performance. The evaluation of these findings was conducted empirically, and theoretical analysis indicates that the quantification of the performance limitation is induced by a low-rank linear language modeling head.

**Questions To Authors:**

What would be the immediate impact of the study and analysis?

**Reasons To Accept:**

The paper is well written for a knowledgable reader. It is difficult to answer a `why' question and the paper showcases a good attempt to characterise a learning problem and provides empirical justifications and theoretical analysis.

**Reasons To Reject:**

It is very difficult to provide an answer/justification to `why' question with language modelling that can be generalisable. But that is the nature of the artificial neural network empirical research paradigm. We still don't know why small models worked well on some problems and did much worse on other problems. Even when it worked well on some problems, it might not have worked well for every instance of that problem.

---

> ### Author Rebuttal · Authors · 2024-05-31
>
> We are grateful to reviewer sGwj for its time and effort, and for raising interesting points about generalization.
>
> ### *Answering the why question in a general way*
> Although our paper has a strong empirical stance, we provide several elements in an attempt to prove the general nature of our observations:
> - An extensive analysis of the intrinsic rank of contextual probability manifolds (Figure 7), that allows an estimation of the dimensionality of target probability distributions in several natural language scenarii.
> - Rank-constrained experiments, performed on several models of different sizes and initially trained on different data (Figure 6).
> - A theoretical exploration (section 5.2) that proves that the connection between the softmax bottleneck and the achievable performance affects all language models trained using cross-entropy on a linear logit language modeling head, especially in the low-dimensional case.
>
> ### *What would be the immediate impact of the study and analysis?*
> There can be several immediate impacts, among others:
> - Including last-layer anisotropy monitoring when training language models to detect saturation;
> - Further exploration of mitigation strategies for the softmax bottleneck issue, specifically designed for language models, with potential performance benefits;
> - Incentivizing the use of larger hidden dimension choices.

---

### Official Review · Reviewer_G2Lq · 2024-05-12

**Rating:** 5
**Confidence:** 3
**Ethics Flag:** 1

**Summary:**

This paper studies relevant properties that lead to performance saturation in small language models. A particular attention is guided towards the rank of linear layer. Through experiments and some theoretical justification, the rank of the linear layer casts an impact on explaining the limited performance of small language models.

**Questions To Authors:**

Dimension of $W$ in Lemmas 5.1, 5.2 and Theorem 5.3 does not match with previous definitions.

Lemma 5.1 seems to be the definition of a global minimizer. Why there is the limit of $\epsilon \to 0$ in Lemma 5.2 (resp. $d \to D$ in Theorem 5.3)? Isn't Lemma 5.2 simply the continuity of the loss function?

**Reasons To Accept:**

The study here is self-contained and explanatory with both empirical and theoretical evidences.

Presentation is relatively good.

As discussed in the related work, the discovery of a tie between the softmax bottleneck to the performance saturation is new to the community.

**Reasons To Reject:**

This study seems to explain a phenomenon, while the explanation does not lead to new methodology design. The discussion at the end of the paper is rather weak on the encouragement of more expressive softmax alternatives.

The theory seems not to add new insights about the phenomenon and the theory itself is very standard in a simplified model. The proof is not very rigorous though, especially the usage of small o and big O notations is not well explained.

Presentation can be improved by providing a clearer connection between Sections 3 - 5.

---

> ### Author Rebuttal · Authors · 2024-05-31
>
> We sincerely thank the reviewer for its highly valuable comments, and especially for its care about the theoretical aspects of the article.
>
> ### 1. *The explanation does not lead to new methodology design*
> Our paper is framed as an extensive study of the connection between the softmax bottleneck and the representation degeneration issue. The drawn conclusions show that there is, to a certain extent, an intrinsic limitation to the performance a language model using a small hidden dimension can achieve. Designing a novel methodology to go around this limitation is definitely a path future works should explore, but as shown in the literature, it represents a difficult problem that probably could lead to a full paper on the subject if such a method was designed. We believe that describing this limitation in depth can pave the way for such methods, and represents an interesting contribution as such.
>
> ### 2. *The theory seems not to add new insights about the phenomenon*
> The goal of the theoretical section is to show that the connection between both phenomena is theoretically grounded. Hence, while we agree that we get no novel insight from this section, we argue that it strengthens our argument and shows its general nature.
>
> ### 3. Mathematical issues
> In a revised version of the paper, we adjusted and corrected some parts of the theoretical part. Notably, we removed lemma 5.1 as it is not necessary for the conclusion, we unified the notations and explained them in details from the start of the article.
>
> Lemma 5.2 can indeed be proven by invoking the smoothness of the loss function, likely by using the $C^1$ nature of cross-entropy to argue that it is Lipschitz continuous, as regular continuity seems insufficient e.g. if the gradient in $W^*$ is undefined. We will add a remark in the proof to underline this. However, our proof relies on limited development, as we wanted to display a more precise relation between the loss gap and the error on the W approximation, and its quasi-linear nature. This formulation will hopefully pave the way for additional exploration of this relation in the future.

---

> > ### Comment · Reviewer_G2Lq · 2024-06-07
> >
> > Thanks for the detailed response. I agree with the discussion on new methodology design, which goes beyond the current scope. This is counted as a limitation, but not to devalue the paper. I believe after a revision on the theoretical part, the paper should be on the bar of publication. I am keeping my initial rating, but would be happy to see the paper being accepted.

---

### Decision · Program_Chairs · 2024-07-10

**Decision:**

Accept

**Comment:**

All reviewers highlight that this paper makes an empirical and theoretical account linking {correlation between loss saturation, anisotropy, effective low rank of the embedding matrix}. There was some concern about clarity and "actionability" which (to my reading) authors adequately addressed in rebuttal period. In short, although this doesn't propose new methodologies, it identifies a theoretical and empirical problem (as well as showing ways for identifying it in models), which opens up a field a future innovations.